# Prefrontal Cortex Cytosolic Proteome and Machine Learning-Based Predictors of Resilience toward Chronic Social Isolation in Rats

**DOI:** 10.3390/ijms25053026

**Published:** 2024-03-06

**Authors:** Dragana Filipović, Božidar Novak, Jinqiu Xiao, Predrag Tadić, Christoph W. Turck

**Affiliations:** 1Department of Molecular Biology and Endocrinology, “VINČA” Institute of Nuclear Sciences, National Institute of the Republic of Serbia, University of Belgrade, 11000 Belgrade, Serbia; 2Proteomics and Biomarkers, Max Planck Institute for Psychiatry, 80804 Munich, Germany; novak@psych.mpg.de (B.N.); turck@psych.mpg.de (C.W.T.); 3Max Planck Institute of Biochemistry, 82152 Martinsried, Germany; jxiao@biochem.mpg.de; 4School of Electrical Engineering, University of Belgrade, 11000 Belgrade, Serbia; ptadic@etf.bg.ac.rs; 5Key Laboratory of Animal Models and Human Disease Mechanisms of Yunnan Province, Kunming Institute of Zoology, Chinese Academy of Sciences, Kunming 650223, China; 6KIZ/CUHK Joint Laboratory of Bioresources and Molecular Research in Common Diseases, Kunming Institute of Zoology, Chinese Academy of Sciences, Kunming 650223, China; 7National Resource Center for Non-Human Primates, National Research Facility for Phenotypic & Genetic Analysis of Model Animals (Primate Facility), Kunming Institute of Zoology, Chinese Academy of Sciences, Kunming 650107, China

**Keywords:** chronic social isolation, resilience, prefrontal cortex, proteomics, machine learning algorithms

## Abstract

Chronic social isolation (CSIS) generates two stress-related phenotypes: resilience and susceptibility. However, the molecular mechanisms underlying CSIS resilience remain unclear. We identified altered proteome components and biochemical pathways and processes in the prefrontal cortex cytosolic fraction in CSIS-resilient rats compared to CSIS-susceptible and control rats using liquid chromatography coupled with tandem mass spectrometry followed by label-free quantification and STRING bioinformatics. A sucrose preference test was performed to distinguish rat phenotypes. Potential predictive proteins discriminating between the CSIS-resilient and CSIS-susceptible groups were identified using machine learning (ML) algorithms: support vector machine-based sequential feature selection and random forest-based feature importance scores. Predominantly, decreased levels of some glycolytic enzymes, G protein-coupled receptor proteins, the Ras subfamily of GTPases proteins, and antioxidant proteins were found in the CSIS-resilient vs. CSIS-susceptible groups. Altered levels of Gapdh, microtubular, cytoskeletal, and calcium-binding proteins were identified between the two phenotypes. Increased levels of proteins involved in GABA synthesis, the proteasome system, nitrogen metabolism, and chaperone-mediated protein folding were identified. Predictive proteins make CSIS-resilient vs. CSIS-susceptible groups linearly separable, whereby a 100% validation accuracy was achieved by ML models. The overall ratio of significantly up- and downregulated cytosolic proteins suggests adaptive cellular alterations as part of the stress-coping process specific for the CSIS-resilient phenotype.

## 1. Introduction

Chronic stress causes biochemical and behavioral reactions in humans, which increase the possibility of developing Major Depressive Disorder (MDD) [1]. Of particular interest are those stressors with a psychosocial component. One of the most commonly used stressors is chronic social isolation (CSIS), an extension of mild chronic stress [2,3]. It is characterized by a disconnection from other social species and a voluntary (actually or seemingly) withdrawal from the social environment [4]. Previous data have shown that changes in the energy metabolism-related proteome are linked to MDD [5,6]. Moreover, analyses of blood and urine samples from depressed individuals have revealed changes in the levels of metabolites that regulate energy metabolism and brain function [7]. Although some individuals are sensitive to stress and have a high risk of developing MDD in response to mild stressors, others are resilient to stress and do not have symptoms in the face of severe adversities [8,9]. The resilient phenotype is a phenomenon that represents a distinct active neurobiological process and not simply the absence of vulnerability [10].

Biological differences between resilience and susceptibility to CSIS can be explored with the help of animal models. CSIS in rats, which are separated from their social environment and lack social interaction, induces depressive-like behavior and evokes a variety of neurochemical and neuroendocrine changes similar to those observed in depressed patients [11,12,13]. In our previous research, the prefrontal cortex (PFC) proteome profiling of adult male rats was used to explore altered cytosolic proteins related to CSIS-induced depression-like behaviors compared to controls [14]. Moreover, differences in the rat hippocampal synaptoproteome profiles between the resilient and susceptible phenotypes toward CSIS suggest modulated physiological responses [15]. Consistent with this, CSIS-resilient rats compared to CSIS-susceptible rats have exhibited downregulated levels of some glycolysis enzymes along with simultaneously upregulated levels of the tricarboxylic acid enzyme (Aco2) and electron transport chain components (Uqcrc2, Atp5f1a, and Atp5f1b), revealing an energy metabolic shift from glycolysis to oxidative phosphorylation in non-synaptic mitochondria in the rat hippocampus [16]. Also, the glutamatergic, serotonergic, and GABA (gamma-aminobutyric acid)ergic systems in brain regions are linked to the resilience vs. susceptibility to social stress [17].

To further investigate new molecular pathways associated with resilience toward CSIS, non-hypothesis-driven proteomic analyses were performed to identify state-specific molecular signatures. Proteomics is capable of revealing the smallest alterations in the proteome profile of a particular condition. Machine learning (ML) algorithms can derive implicit patterns from large biological datasets and have been applied to identify predictive proteins for each condition. We applied quantitative proteomics using liquid chromatography–tandem mass spectrometry (LC-MS/MS) analysis followed by label-free quantification and STRING bioinformatics combined with class-separation and ML algorithms such as a support vector machine (SVM) with sequential feature selection and random forest (RF) classifiers. The main goal was to identify altered proteome components along with biochemical pathways and processes as well as potential predictive proteins specific for CSIS-resilient rats relative to CSIS-susceptible and control rats by analyzing the global protein expression in the cytosolic-enriched fraction of the PFC. We focused on the cytosol fraction, because it is involved in a wide range of basic biochemical processes, such as glycolysis [18], the pentose phosphate pathway [19], and protein synthesis and degradation, that control the level of protein expression [20], signal transduction [21], stress response signaling [22], and antioxidative enzymes [23]. We profiled differences in the cytosolic proteome using a time course representative of the events underlying the development of resilience or susceptibility to CSIS. PFC was chosen as a sensitive brain region to stress that participates in cognitive processes and socio-emotional functions [24]. Moreover, a reduced volume and dendritic spine density of the PFC have been detected in depressed patients and in experimental animals that underwent the chronic stress paradigm [25]. In chronic stress conditions, the PFC undergoes significant physiological changes to cope with the demands associated with cellular activation.

Since stress resilience is a common clinical phenomenon, the inclusion of an unsusceptible group to CSIS increases the usefulness of the model and provides important data for translational resilience research. Knowledge of the neural mechanisms underlying stress resilience may enable the successful treatment of stress-related psychiatric disorders [17]. To our knowledge, this is the first study investigating the PFC cytosolic proteome combined with the ML-driven identification of predictive proteins for CSIS resiliency.

## 2. Results

### 2.1. Behavioral Assessment of CSIS Rats

After the 6-week CSIS treatment, the adult male *Wistar* rats were divided into two stress subtypes based on the behavioral testing data. These CSIS rats were assigned to CSIS-susceptible (anhedonic-like) when their decrease in sucrose intake was ≥30% and CSIS-resilient (resilient to CSIS-induced anhedonia) when their sucrose intake was not significantly different from that of the baseline [16]. The results of the sucrose preference test (SPT) are shown in Figure 1. Repeated ANOVA revealed a significant main effect of CSIS (F_2,42_ = 15.06, *p* < 0.001) and significant interactions between time and CSIS (F_4,42_ = 7.25, *p* < 0.001). The sucrose preference (SP) data showed a decrease in SP at the end of the 3rd and 6th weeks of CSIS compared to that at the baseline (0 weeks) (* *p* < 0.05, *** *p* < 0.001), which is indicative of anhedonic- or depressive-like behavior (CSIS-susceptible rats). No behavioral distinction in terms of the SP at the end of the 3rd week or 6th week compared to that at the baseline was found in the CSIS-resilient rats.

### 2.2. Comparative Protein Analysis of the Cytosolic-Enriched Fraction of Rat PFC

In comparing the cytosolic proteomes of the CSIS-resilient and CSIS-susceptible groups, a total of 1409 and 1090 proteins were identified and quantified, respectively. Based on a fold change (F.C) ≤ 0.80 and FC ≥ 1.2, with an FDR-corrected *p* < 0.05, a total of 367 significantly differentially expressed proteins were identified, of which 165 differential proteins were downregulated and 202 upregulated (Appendix A). When comparing the CSIS-resilient and control groups, a total of 1389 and 1123 proteins were identified and quantified, respectively, whereby only one upregulated protein was found (adjusted *p*-value (BH) < 0.05 and FCs < 0.8 and >1.2) (Appendix A).

Principal component analysis (PCA) was performed to reflect the protein differences among samples and the variation between samples in the group. The 2D PCA score plot is presented in Figure 2A,B. It is defined by two principal components: PC1 (64.8%) and PC2 (11%) for CSIS-resilient vs. CSIS-susceptible rats, and PC1 (37.4%) and PC2 (19.5%) for the CSIS-resilient vs. control groups. The PCA results revealed that the CSIS-resilient samples were separated from those of the CSIS-susceptible group, while the samples from the CSIS-susceptible group were not distinguishable from the control samples. Protein expression differences between the control rats and CSIS-treatment rats are also presented in a volcano map (Figure 2C,D).

### 2.3. Class-Separating Proteins

A panel of the top 15 cytosolic potential predictive proteins for discriminating CSIS-resilient vs. CSIS-susceptible rats is presented in Table 1. All proteins obtained by class separation are shown from the highest to the lowest score (Appendix A).

### 2.4. Network Analysis of Protein–Protein Interactions (PPIs)

Cytosolic predictive proteins (n = 45) discriminating CSIS-resilient vs. CSIS-susceptible rats obtained by class separation were analyzed with the STRING 11.0 software to predict protein–protein interaction (PPIs). The gene ontology (GO) annotation included a biological process, molecular function, and KEGG pathway. An interactome study showed significant interactions between proteins with an enrichment *p*-value of 4.88 × 10^−15^. At the biological process gene ontology (GO) level, 14 biological processes were found, whereby predictive proteins were mostly involved in protein folding (GO:0006457), the response to nitrogen (GO:1901698), and oxygen-containing compounds (GO:1901701). The most affected molecular functions (n = 26) were G protein-coupled receptor binding (GO:0001664), GDP/GTP binding (GO:0019003, GO:0005525), and signaling receptor binding (GO:0005102). KEGG pathway enrichment (n = 58) indicated that the identified proteins are mainly involved in tyrosine metabolism (rno00350), the serotonergic synapse (rno04726), GABAergic synapses (rno04727), the cAMP signaling pathway (rno04024), carbon metabolism (rno01200), and glutamatergic synapses (rno04724) (Figure 3).

### 2.5. SVM with Greedy Forward Search and RF Classification

Common cytosolic predictive proteins for classifying CSIS-resilient vs. CSIS-susceptible rats using SVM-based sequential feature selection and an RF are presented in Table 2. Potential predictive proteins are ranked by the number of times they were selected in the 100 repetitions. The full list of predictive proteins obtained through these classifications is presented in Appendix A.

## 3. Discussion

This study identified altered proteome components and biochemical pathways and processes in the PFC cytosolic-enriched fraction in CSIS-resilient rats compared to CSIS-susceptible and control rats. The identification of potential predictive proteins was performed using class separators. Additional predictive proteins were identified through an SVM-based sequential feature selection and RF-based feature importance.

CSIS-resilient rats displayed behavioral differences in terms of SP when compared to the CSIS-susceptible group and showed significant cytosolic proteome changes. Among the identified proteins, reduced levels of proteins involved in glycolytic and pyruvate metabolic processes, such as Hk1, Aldoa, Aldoc, Pgk1, Pkm, and Ldhb, were found in CSIS-resilient rats compared to CSIS-susceptible rats. Pkm is a glycolytic enzyme that converts phosphoenolpyruvate to pyruvate in a reversible reaction at the endpoint of the glycolytic pathway, and its modulation determines the glucose flux. Its decreased levels probably reduce the glycolytic flux, leading to a reduction in the production of pyruvate. The same trend was observed for Ldhb, which catalyzes the reversible conversion of pyruvate to lactate as well as NAD+ to NADH. Accordingly, Hk1, Aldoa, Aldoc, and Pgk1 suggest the already-mentioned decreased energy demands of CSIS-resilient rats. These data are in line with our previous study’s results showing a decline in glucose metabolism in the hippocampus of CSIS-resilient rats [16]. Conversely, abnormal brain energy metabolism has been found to be a contributing factor for MDD. Pkm and Ldh were significantly upregulated in the cerebella from chronic, mildly stressed rats, indicating an activated glycolytic process [26]. A proteomic study on the hypothalamus samples of CUMS mice showed an upregulation in Hk1 [27]. Decreased levels of Hk1 and Aldoa in the current study were also identified as potential predictive proteins contributing to the CSIS resiliency designation by the RF or class separation. Accordingly, HK1 and Aldo may be potential treatment targets. Interestingly, increased levels of the multifunctional protein Gapdh (Appendix A) were observed in CSIS-resilient rats. A previous proteomic study with the cerebrospinal fluid of a monkey model of depression and MDD patients, indicated a downregulated Gapdh expression [28]. Apart from its key role in glycolysis [29], Gapdh also displays activities related to the regulation of microtubule binding and cytoskeletal dynamics, the modulation of the calcium current component of AMPARs, and inositol trisphosphate (IP3) receptors [30]. Moreover, its pathological or physiological function depends on post-translational modifications [31]. Indeed, we found an upregulation in microtubule-associated proteins (Map6, Map2, Map1b, Map4, Map1a), cytoskeleton organization (Tubb4b, Tuba1a, Tubb2s, Tubb3 Tubb2b, Tuba4a, Tubb5) and calcium/calmodulin-dependent protein kinase (Camk1, Camkk1, Camk4, Camkv, Camkk2). The upregulation in the cytoskeletal microtubular system may imply increased microtubule dynamics. Increased levels of calcium binding proteins likely enable compensation for increased calcium fluxes within neurons in the PFC. Also, Tubb3 and Camkk2 were identified as potential predictive proteins according to the RF. In addition, an increased level of Exportin-1, a protein responsible for nuclear protein export, which is also associated with Gapdh translocation [32], was found. The upregulations in these proteins in CSIS-resilient rats likely reflect a positive molecular path for coping with the CSIS.

Decreased levels of Gnai1 and Gnai2 proteins, which participate in the G protein-coupled receptor, were also identified as predictive proteins via class separation. Moreover, these differentially expressed proteins are involved in different KEGG pathways (Figure 3). Given that the phosphatidylinositol signaling system mainly comprises receptors and G proteins, the lower levels of Gnai1 and Gnai2 in CSIS-resilient rats compared to CSIS-susceptible rats may contribute to an altered phosphatidylinositol signaling pathway. In support of this, downregulated Impa1, responsible for the production of inositol and essential for the synthesis of phosphatidylinositol, was found. Moreover, a recent metabolomic study demonstrated increased levels of the components of phosphatidylinositol, such as myo-inositol, following CSIS (referred to as CSIS-susceptible rats) in the PFC as one of the most interesting variables for stratifying CSIS-susceptible vs. control rats [33]. Downregulated proteins involved in the Ras subfamily of GTPases, such as Rap1b, Rab1A, Rab2a, and Rab1b, were identified as predictive proteins using ML algorithms. As these proteins are involved in membrane trafficking, neurotransmitter release, and signal transduction [34], their downregulation may cause a decrease in neuronal vesicular cell trafficking. We also identified the downregulation of Cycs, which was found as a predictive protein via class separation. We recently reported an increase in cytochrome c levels in the cytosol of the PFC in CSIS-susceptible rats [14], which may suggest a compromised mitochondrial membrane integrity and its concomitant release from mitochondria into the cytosol. Moreover, two class-separated predictive proteins, Ncam and Capns, with opposite expression differences were found in the CSIS-resilient rats compared to the CSIS-susceptible rats. Ncam, as a cell surface communication protein, is known to participate in the formation of neuronal networks in the brain [35]. Its lower abundance may indicate perturbed cell-to-cell interactions, including alterations in the synaptic plasticity processes. Contrary to this, cytosolic calcium-dependent proteases, such as Capns, were upregulated. Capns is involved in synaptic plasticity through the regulation of protein targets and signaling pathways, with important roles in long-term potentiation [36]. Its increased abundance might indicate a neuronal protection mechanism as part of a stress-coping process specific for the CSIS-resilient phenotype. 

The energy metabolism of neurons and astrocytes is related to the synthesis and metabolism of glutamate and GABA, as excitatory and inhibitory neurotransmitters of the brain, respectively [37]. The biosynthesis and turnover of glutamate and GABA are largely dependent on astrocytes, since they are the only cells expressing enzymes required for the de novo synthesis of the two amino acids. Here, we identified increased levels of Glul in CSIS-resilient vs. CSIS-susceptible rats. Given that this enzyme converts glutamate and ammonia to glutamine in astrocytes [38], we can assume that upregulating Glul in cells counteracts the cytotoxicity of this neurotransmitter. Moreover, the upregulation of predictive protein Gfap (SVM-based sequential feature selection), an astrocyte-specific cytoskeletal protein involved in the regulation of glutamate transporter trafficking and function, may indicate increased glutamate levels in the brain. Gfap has been found to promote glutamate aspartate transporter association with the plasma membrane by forming a complex with the cytoskeleton-associated linker protein ezrin [39], also found as a predictive protein. These results point to the glutamate-modulating role of astrocytes as being important for CSIS resilience Clinical studies have found reduced Gfap levels in depressed subjects in cortical regions [40], and in a chronic mild stress-anhedonic group compared to a resilient group [41]. Moreover, glial loss may, in part, underlie the cognitive symptoms of depression, pointing to a glial pathology hypothesis of depression [42]. In addition, increased levels of Gad1 and Gad2, which regulate GABA synthesis from glutamate, probably contributed to the normalization of GABA- and glutamate-mediated neurotransmission and were deregulated in CSIS-susceptible rats. Reduced GABA neurotransmission has been observed in MDD patients [43]. We previously reported the dysfunction in the GABAergic system in the PFC, specifically of the parvalbumin-expressing interneurons, as a response to CSIS exposure [44].

Proteins that were upregulated in CSIS-resilient vs. CSIS-susceptible rats were related to proteasome systems. Given that the proteasome system is responsible for the degradation of short-lived, misfolded, and damaged proteins, we presume that upregulated proteasome components, such as Psmb9, Psmb2, Psmb13, Psmb1, and Psmb11, prevent the accumulation of misfolded and modified proteins. Our previous PFC subproteome of CSIS-susceptible rats compared to control rats showed decreased levels of the cytosolic proteins involved in the proteasome system [14]. Observed differences in the proteasome’s protein levels may be explained by a higher turnover of cytosolic proteins in CSIS-resilient rats compared to CSIS-susceptible rats. Changes in the levels of several regulatory proteins indicate the modulation of cell signaling pathways. Hence, upregulated protein levels of potential predictive Rab18 (SVM-based sequential feature selection) and Arfgap1 stimulate the process of intracellular membrane trafficking. Among the interesting differentially upregulated proteins were heat shock proteins Hspa2 and Hsph1, which were found as potential predictive proteins through the RF. They ensure the correct folding and re-folding of misfolded proteins, indicating the increased dynamics and expressions of proteins. However, potential predictive chaperones with the same role, such as Hspa5, Hspa9, and Hsp90b1 (via class separation or FR), as well as Fkbp4, showed a lower abundance in CSIS-resilient vs. CSIS-susceptible rats. This result implicates the redirection of heat shock proteins to increased levels of Hspa2 and Hspb1 toward resisting the adverse effects of CSIS. Nonetheless, the lower abundances of potential predictive proteins Cat and Gpx1 via SVM-based sequential feature selection, involved in the antioxidative system, may suggest that CSIS-resilient rats are less challenged by oxidative stress. Given that Cat and Gpx1 maintain cellular redox balance by regulating reactive oxygen species levels, these proteins may be potential MDD therapeutic targets. Regarding CSIS-resilient compared to control rats, only minor proteome changes were observed. Upregulation was observed for Fah, the enzyme that catalyzes the last step of tyrosine catabolism. In addition, this protein was downregulated in CSIS-resilient rats compared to CSIS-susceptible rats. The perturbation of Fah protein levels is likely due to physiological conditions related to CSIS. We submit that predictive proteins identified with ML algorithms based on proteomic PFC data are able to delineate subjects resilient vs. susceptible to CSIS. Moreover, the predictive proteins of stress resilience may provide information about susceptibility to psychosocial stress [45] and provide new strategies for the prevention and treatment of MDD.

## 4. Materials and Methods

### 4.1. Animals

We used adult male *Wistar* rats (2.5 months of age, 300–350 g in body weight) bred at the Animal Facility of the “VINČA” Institute of Nuclear Sciences, National Institute of the Republic of Serbia, University of Belgrade. The rats were kept under standard conditions in groups of up to four per cage with a 12 h light/dark cycle, a temperature of 20 ± 2 °C, a humidity level of 55 ± 10%, and free access to food and water ad libitum. All experimental procedures were approved by the Ethical Committee for the Use of Laboratory Animals of the “VINČA” Institute of Nuclear Sciences, National Institute of the Republic of Serbia, University of Belgrade, which follows the guidelines of the EU-registered Serbian Laboratory Animal Science Association (SLASA). The study protocol was approved by the Ministry of Agriculture, Forestry, and Water Management—Veterinary Directorate, ethics committee, license numbers 323-07-01893/2015-05 and 323-07-02256/2019-05.

### 4.2. Study Design

At the onset of the experiment, rats were randomly divided into control rats, which were housed in groups of four animals per cage, and rats that underwent CSIS for six weeks, which were housed individually in a cage and deprived of any visual or tactile contact with other animals, but had normal auditory and olfactory experiences [46]. Following the six-week period of CSIS, the rats were designated CSIS-resilient and CSIS-susceptible based on their performance in the SPT, as previously described [16] (Figure 4).

#### Sucrose Preference Test (SPT)

The SPT was employed to segregate CSIS-resilient and CSIS-susceptible rats, according to the method described previously [16]. The rats were placed into separate cages and two pre-weighed bottles, one containing tap water and the other containing a 2% sucrose solution, were placed on each cage. We measured the percentage preference of the rats for 2% sucrose solution compared to tap water after 1 h. A decline in SP was indicative of anhedonia or a decreased ability to experience pleasure, a major symptom of depression [47]. The test was conducted prior to the start (baseline, week 0) and at the end of the 3rd and 6th weeks.

### 4.3. Isolation of PFC Cytosolic-Enriched Fractions

Once behavioral tests were completed, the control (n = 8), CSIS-susceptible (n = 8), and CSIS-resilient (n = 8) rats were anesthetized with intraperitoneal injections of ketamine/xylazine (100/10 mg/kg) (Richter Pharma AG, Wels, Austria/Bioveta, Ivanovice na Hane, Czech Republic), perfused with physiological saline, and sacrificed via guillotine (Harvard Apparatus, South Natick, MA, USA) decapitation. The brain was removed, and the PFC excised on ice, shock-frozen in liquid nitrogen, and stored at −80 °C until further analyses. To obtain cytosolic-enriched fractions, the PFC was homogenized in 1 mL of cold homogenization buffer (10 mM Tris/HCl (SERVA) pH 7.4, 0.25 M sucrose (Thermo Fisher Scientific, Waltham, MA, USA) containing a protease inhibitor cocktail tablet (cOmplete™, EDTA-free Protease Inhibitor Cocktail No 4693132001) [14]. Homogenization was performed in a Potter–Elvehjem homogenizer (800 rpm, 12 up-and-down passes). Homogenates were centrifuged at 1000× *g* for 10 min to obtain the crude nuclear pellet (P1) and the supernatant (S1). The S1 fraction was again centrifuged under the same conditions for the removal of the remaining nuclei. The obtained supernatant was then centrifuged for 15 min at 17,000× *g* to obtain the crude mitochondrial fraction (P2 pellet), and supernatant S2 was centrifuged at 100,000× *g* at 4 °C for 60 min to obtain cytosolic-enriched fractions. Protein concentrations were measured using the Lowry method [48] and the bicinchoninic acid method (Pierce™ BCA Protein Assay Kit, Thermo Scientific, Waltham, MA, USA), using purified BSA (Sigma-Aldrich, Munich, Germany) as a standard. The relative purity of isolated cytosolic-enriched fractions was confirmed by the absence of nuclear/mitochondrial contaminants of the cytosolic fractions after the incubation of control samples with antibody against nuclear (anti-TATA binding protein), cytosolic (anti-α tubulin), and mitochondrial (anti-voltage-dependent anion channel 1) proteins, as described in our previous study [14]. 

### 4.4. Enzymatic Digestion of Proteins and Liquid Chromatography Coupled with Tandem Mass Spectrometry Analysis (LC-MS/MS)

The cytosolic proteins were digested following the filter-aided sample preparation method [49] with minor modifications, as described previously [14]. Briefly, 20 µg of protein from cytosolic-enriched fractions was diluted with urea buffer (8 M urea, 100 mM Tris-HCl pH 8.5) (Sigma-Aldrich, Munich, Germany) to a total volume of 100 µL containing 5 µL of 0.1 M DTT (final concentration, 5 mM), transferred to a Microcon YM-30 filter device (Merck Millipore, Darmstadt, Germany), incubated for 1 h at 37 °C in a thermo-mixer (300 rpm) and centrifuged at 14,000× *g* for 15 min at room temperature (RT). Then, 100 µL of urea solution was added to the samples and centrifuged for 15 min, and this step was repeated three times. Then, 100 µL of urea buffer containing 5.25 µL of iodoacetamide (IAA) (Sigma-Aldrich, Munich, Germany) (final concentration, 10 mM) was added onto the filter. The samples were incubated for 45 min at 24 °C in the dark with mixing (300 rpm) and centrifuging for 15 min at 14,000× *g* at RT. Next, the samples were washed three times with 150 µL 100 mM triethylammonium bicarbonate (TEAB), pH 8.5 (dissolution buffer) (Thermo Scientific, Waltham, MA, USA). Proteins were digested overnight at 37 °C in 150 µL 100 mM TEAB with 0.5 µg trypsin (Trypsin Premium Grade) (trypsin-to-protein ratio = 1:50) (Serva). Peptides were centrifuged for 15 min at 14,000× *g* and washed with 150 µL of dissolution buffer. The reaction was terminated by adding 1 µL of formic acid (FA) (Thermo Fisher Scientific, Waltham, MA, USA) to the 100 µL solution, and the samples were desalted with Pierce C18 10 µL tips (Thermo Scientific, Sunnyvale, CA, USA) following the manufacturer’s instructions and dried in a speed-vac. The peptides resuspended in 0.1% FA were analyzed via LC-MS/MS using a Dionex Ultimate 3000 RSLC nano UPLC (Thermo Fisher Scientific, Waltham, MA, USA) system coupled to a Q Exactive Plus mass spectrometer (Thermo Fisher Scientific, Waltham, MA, USA), as previously published [14]. All *m*/*z* values of the eluting precursor ions were measured in an Orbitrap mass analyzer, set at a resolution of 70,000. Data-dependent scans (Top 10) were employed to automatically isolate and generate fragment ions via high energy collision dissociation (HCD) in the quadrupole mass analyzer. The resulting fragment ions were measured in the Orbitrap analyzer, set at a resolution of 17,500. Peptide ions with charge states of 2+ to 4+ were selected for fragmentation [50].

### 4.5. Data Processing for Label-Free Quantification

The mass spectrometry raw data were analyzed with MaxQuant, version 1.6.3.4 [51,52]. For identification, the rat reference proteome from UniProt (downloaded 22 December 2020) was used as a reference database. Enzyme specificity was set to trypsin, allowing N-terminal cleavage before proline. Variable modifications were set to the oxidation of methionine residues and acetylation of protein N termini, whereas the carbamidomethylation of cysteine residues was set as a fixed modification. A maximum of two missed cleavages was allowed. A false discovery rate (FDR) of 1% was used for peptide and protein identification. Peptide identification was based on a search with an initial mass deviation of the precursor ion of up to 10 ppm. The fragment mass tolerance was set to 0.1 Da on the *m/z* scale. Only proteins identified with at least two peptides were considered for relative quantification. The mass spectrometry proteomics data were deposited to the ProteomeXchangeConsortium through the PRIDE [53] partner repository with the dataset identifier PXD048641.

### 4.6. Statistical and Bioinformatic Analysis

The statistical significance of the mass spectrometry label free data was determined using a two-tailed unpaired t-test followed by an appropriate false-discovery rate (FDR q < 0.05) correction using the Benjamin–Hochberg method. Only proteins showing protein FC greater than or equal to 1.2 (F.C. ≥ 1.2) or less than or equal to 0.80 (F.C ≤ 0.80), and a *p* value < 0.05 and FDR < 0.05 were considered differentially expressed. Proteins and peptides identified according to only one peptide match and/or one unique peptide were excluded from the bioinformatic analysis. The STRING online tool (https://string-db.org/), accessed on 25 December 2023, was used to identify protein–protein interaction networks on potential predictive proteins between CSIS-resilient and CSIS-susceptible rats according to their UniProtKB accession numbers, and to define biological processes, molecular functions, and KEGG pathways. 

### 4.7. Feature Selection with SVM and RF Algorithms

We interpreted the label-free mass spectrometry data as features and defined a classification task for discriminating between CSIS-resilient and CSUS-susceptible rats. Any missing values were replaced using the non-missing values of that same feature. Each feature was standardized by subtracting the mean and scaling to unit variance. We then used two machine learning-based procedures for estimating the predictive power of each feature: (1) sequential feature selection (also known as greedy forward search) and (2) feature importance scores produced as a side-result of training an RF. 

In sequential feature selection (SFS), we started with an empty set and added, one by one, features that result in a classifier with the highest cross-validation accuracy [54]. We employed stratified 7-fold cross-validation to evaluate the classifier performance. The procedure of adding new members into the set of already selected features was repeated until reaching a 100% validation accuracy or when adding new features did not improve the cross-validation accuracy score. The score of a feature was defined as the increase in the cross-validation accuracy brought about by adding that feature into the set of predictors used by the classifier. To break the dependence of our results on the arbitrary ordering of features, we repeated the entire procedure 100 times, randomly shuffling the order in which the features were evaluated in each repetition. We also changed the random seed used to control the distribution of examples over the cross-validation folds in each repetition. We used a support vector machine [55] with a linear kernel and L1 regularization as the base learner. 

The described SFS procedure is of limited use if the dataset contains features that enable perfect classification on their own. We defined a feature as class-separating if its lowest value among all examples of one class was higher than its largest value in the other class. For example, the highest intensity value for protein Q64537 in the CSIS-susceptible group was 10,153,000, while the lowest value for that same protein among the CSIS-resilient individuals was 17,733,000. In other words, the values for Q64537 were much lower for all individuals from the CSIS-susceptible group than for any individual from the CSIS-susceptible group. For such features, the SFS procedure stops after the first iteration, having achieved a 100% validation accuracy by selecting a class-separating feature. To avoid a situation where class-separating features mask other informative, non-class-separating features, we removed the class-separating features from consideration prior to running the SFS procedure. The score of a class-separating feature was computed as the absolute difference between the two nearest (standardized) values corresponding to different classes.

A RF is an ensembles of decision trees, each trained on a bootstrapped dataset using the random subspace method [56]. Each node of a tree is a thresholding test on the value of a single feature. Such tests split the training examples into two groups, which are evaluated in terms of the Gini impurity of the corresponding class labels (if all examples in the group belong to the same class, that group has minimal impurity). To define a single node, we chose a random subset of features (a random subspace) and selected the feature that results in the biggest decrease of impurity. In addition, the average reduction in impurity for a single feature represents a measure of its informativeness. We used these measures as scores to define a ranking of features based on their predictive power for the classification task at hand. We repeated the procedure 100 times, with different seeds for the random number generator that controls the random subspace selection process. We selected the hyper-parameters of our RF by maximizing the out-of-bag (OOB) score, which is a commonly used proxy for validation accuracy [56]. 

Our code was written in the Python programming language. We implemented the procedure for identifying the class-separating features ourselves and used the scikit-learn library [57] for the training of the SVM and RF classifiers.

## 5. Conclusions

We profiled PFC cytosolic proteome changes in CSIS-resilient rats compared to CSIS-susceptible and control rats, providing new insights into the dynamic molecular mechanisms of CSIS resilience. We identified potential predictive proteins that can distinguish the two groups using class separation, SVM-based sequential feature selection, and RF-based feature importance algorithms. Among the different proteomic profilings observed, the glycolysis may be specifically involved in defining resilience to CSIS. Decreased levels of some glycolytic enzymes likely provide less energy in CSIS-resilient rats compared to CSIS-susceptible rats, leading to a reduction in the production of pyruvate, which is consistent with the lower abundance of Pkm and Ldhb proteins. Reductions in the levels of the G protein-coupled receptor, Ras subfamily of GTPases, and antioxidative system were also found in CSIS-resilient vs. CSIS susceptible rats. Conversely, increased levels of Gapdh and proteins involved in microtubule and cytoskeletal organization, calcium-binding proteins, glutamate and GABA metabolic processes, the proteasome system, and chaperone-mediated protein folding were identified between the two phenotypes. The finding of upregulated Fah protein levels in CSIS-resilient vs. control rats, but downregulated Fah levels in CSIS-resilient vs. CSIS-susceptible rats is likely due to physiological conditions related to CSIS rats. Predictive proteins make CSIS-resilient vs. CSIS-susceptible groups linearly separable, whereby a 100% validation accuracy was achieved by the ML models. Overall, proteomic data-driven class separation and ML algorithms can provide a platform for delineating predictive proteins and providing insights into the molecular mechanisms underlying resilience vs. susceptibility to stressful events. 

## Figures and Tables

**Figure 1 ijms-25-03026-f001:**
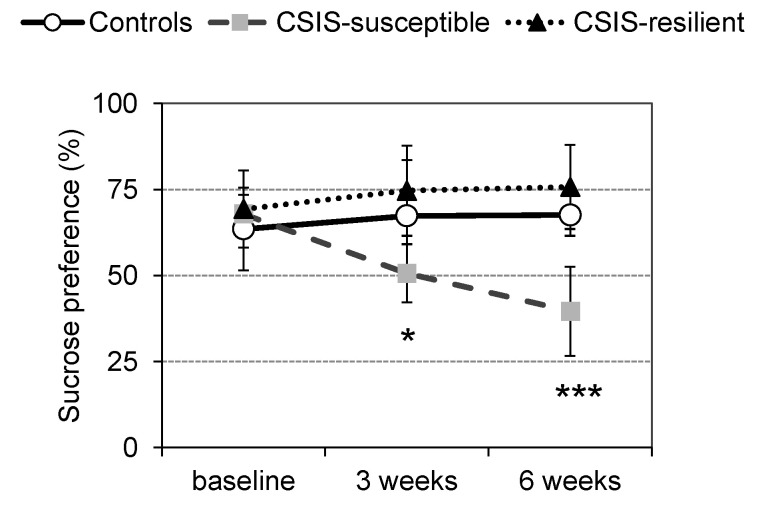
Sucrose preference test (SPT) results of the control, CSIS-susceptible, and CSIS-resilient rats at baseline (0 weeks) and after the 3rd and 6th weeks. Significant differences between groups obtained from repeated ANOVA and post-hoc Duncan’s test are indicated as follows: CSIS-susceptible (3 weeks) vs. CSIS (baseline), * *p* < 0.05; CSIS-susceptible (6 week) vs. CSIS (baseline), *** *p* < 0.001. Data are expressed as the mean ± standard deviation (±SDEV), n = 8 rats per group.

**Figure 2 ijms-25-03026-f002:**
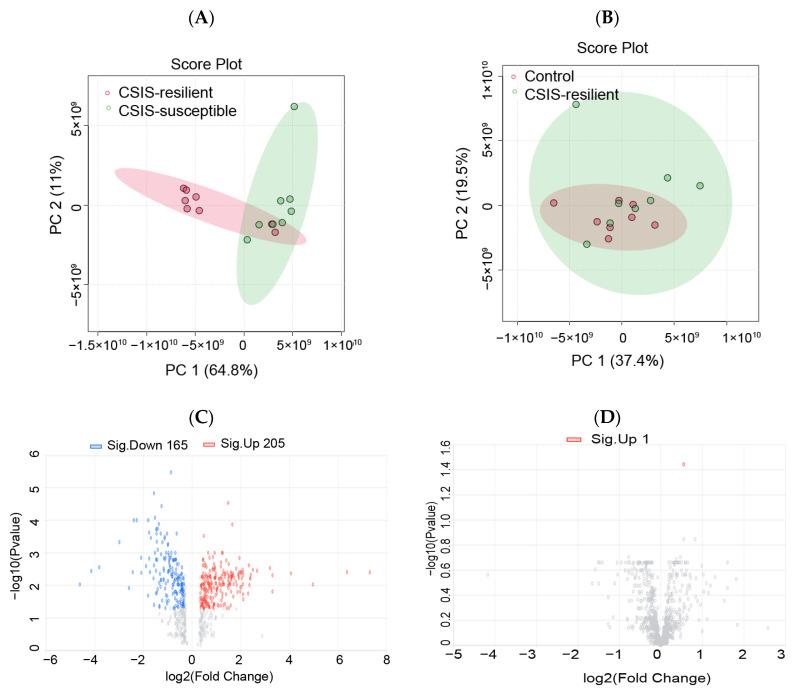
Differential protein analysis from proteomic profiling. The identification of differentially expressed proteins in the prefrontal cortex. Principal component analysis (PCA) of score plot between CSIS-resilient and CSIS susceptible rats (**A**) and CSIS-resilient and control rats (**B**). Volcano plot displaying differentially expressed proteins in the CSIS-resilient vs. CSIS-susceptible rats (**C**), and CSIS-resilient vs. Control rats (**D**) (log (base 2), *x*-axis; negative false log discovery rate (*p*-value) (base 10), *y*-axis). Upregulated proteins are indicated in red, and those downregulated, in blue.

**Figure 3 ijms-25-03026-f003:**
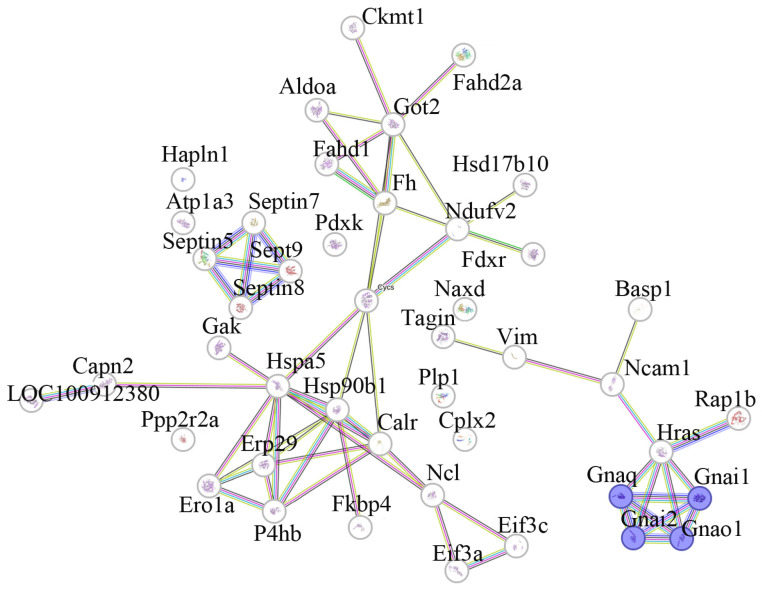
Schematic representation of protein–protein interactions (PPIs) among cytosolic potential predictive proteins between CSIS-resilient and CSIS-susceptible rats, including proteins involved in the glutamatergic synapse KEEG pathway represented in blue. Gnaq—Guanine nucleotide-binding protein G(q) subunit alpha; Gnai2—Guanine nucleotide-binding protein G(i) subunit alpha-2; Gnao1—Guanine nucleotide-binding protein G(o) subunit alpha; Gnai1—Guanine nucleotide-binding protein G(i) subunit alpha-1.

**Figure 4 ijms-25-03026-f004:**
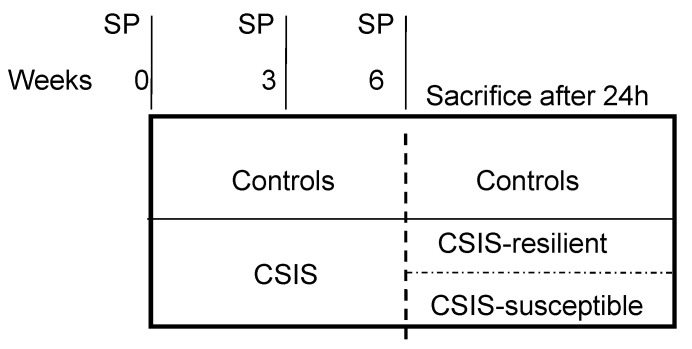
Study design. SP—sucrose preference, CSIS—chronic social isolation, n = 8 rats per group.

**Table 1 ijms-25-03026-t001:** List of top 15 cytosolic proteins discriminating the CSIS-resilient vs. CSIS-susceptible groups with FC (fold change) by class separation.

Protein ID	Protein	Gene	FC
Q64537	Calpain small subunit 1	*Capns1*	3.15
P62898	Cytochrome c, somatic	*Cycs*	0.38
B2RYW9	Fumarylacetoacetate hydrolase domain-containing protein 2	*Fahd2*	0.42
P10824	Guanine nucleotide-binding protein G(i) subunit alpha-1	*Gnai1*	0.36
P05065	Fructose-bisphosphate aldolase A	*Aldoa*	0.55
P14408	Fumarate hydratase, mitochondrial	*Fh*	0.34
P59215	Guanine nucleotide-binding protein G(o) subunit alpha	*Gnao1*	0.36
Q9QVC8	Peptidyl-prolyl cis-trans isomerase FKBP4	*Fkbp4*	0.75
P06761	Endoplasmatic reticulum chaperone BiP	*Hspa5*	0.41
Q62636	Ras-related protein Rap-1b	*Rap1b*	0.33
P60203	Myelin proteolipid protein	*Plp1*	0.06
P31232	Transgelin	*Tagln*	0.57
P13596	Neural cell adhesion molecule 1	*Ncam1*	0.35
P52555	Endoplasmic reticulum resident protein 29	*Erp29*	0.50
P06687	Sodium/potassium-transporting ATPase subunit alpha-3	*Atp1a3*	0.04

**Table 2 ijms-25-03026-t002:** List of cytosolic potential predictive proteins for classifying CSIS-resilient vs. CSIS-susceptible rats identified through SVM-based sequential feature selection and RF importance scores only for those proteins ranked by the number of selections in the 100 repetitions; FC—fold change. Blank cells correspond to proteins not identified as informative by the RF-based procedure in any of the 100 repetitions.

Protein IDs	Protein	Gene	SVM Score	SVM Score SDEV	Times Selected by SVM	RF Score	RF Score SDEV	Times Selected by RF	FC
P61980	Heterogeneous nuclear ribonucleoprotein K	*Hnrnpk*	0.738	0.380	17	0.020	0.008	38	1.39
Q641X8	Eukaryotic translation initiation factor 3 subunit E	*Eif3e*	0.946	0.010	12	0.017	0.007	47	1.82
P47819	Glial fibrillary acidic protein	*Gfap*	0.948	0.010	10	0.015	0.003	18	3.85
P35332	Hippocalcin-like protein 4	*Hpcal4*	0.943	0.012	10	0.016	0.004	19	1.88
P25093	Fumarylacetoacetase	*Fah*	0.949	0.008	7	0.015	0.003	22	0.46
P54313	Guanine nucleotide-binding protein G(I)/G(S)/G(T) subunit beta-2	*Gnb2*	0.946	0.011	7	0.016	0.005	24	0.47
P42676	Neurolysin, mitochondrial	*Nln*	0.816	0.314	7	0.016	0.006	35	0.62
P54290	Voltage-dependent calcium channel subunit alpha-2/delta-1	*Cacna2d1*	0.051	0.008	7	0.016	0.004	31	0.26
Q794E4	Heterogeneous nuclear ribonucleoprotein F	*Hnrnpf*	0.500	0.452	6	0.017	0.007	30	1.37
Q6NYB7	Ras-related protein Rab-1A	*Rab1A*	0.952	0.000	5	0.017	0.007	30	0.50
Q1HCL7	NAD kinase 2, mitochondrial	*Nadk2*	0.948	0.010	5	0.016	0.004	21	0.53
P62994	Growth factor receptor-bound protein 2	*Grb2*	0.062	0.012	5	0.017	0.008	11	1.43
P84076	Neuron-specific calcium-binding protein hippocalcin	*Hpca*	0.052	0.010	5				1.48
O35180	Endophilin-A3	*Sh3gl3*	0.052	0.010	5				1.39
Q80Z29	Nicotinamide phosphoribosyl transferase	*Nampt*	0.952	0.000	4	0.016	0.004	19	0.78
Q68FY0	Cytochrome b-c1 complex subunit 1, mitochondrial	*Uqcrc1*	0.952	0.000	4	0.016	0.004	19	0.34
P01830	Thy-1 membrane glycoprotein	*Thy1*	0.280	0.388	4	0.018	0.006	29	0.14
Q6P686	Osteoclast-stimulating factor 1	*Ostf1*	0.054	0.010	4				0.53
Q5FVI6	V-type proton ATPase subunit C 1	*Atp6v1c1*	0.054	0.010	4				0.74
P07340	Sodium/potassium-transporting ATPase subunit beta-1	*Atp1b1*	0.054	0.010	4				0.09
Q704S8	Carnitine O-acetyltransferase	*Crat*	0.048	0.000	4				0.71
P32736	Opioid-binding protein/cell adhesion molecule	*Opcml*	0.048	0.000	4	0.017	0.007	27	0.33
Q62703	Reticulocalbin-2	*Rcn2*	0.944	0.011	3	0.018	0.007	38	0.51
P70566	Tropomodulin-2	*Tmod2*	0.056	0.011	3	0.017	0.007	38	1.90
Q66X93	Staphylococcal nuclease domain-containing protein 1	*Snd1*	0.048	0.000	3				0.79
Q9QXQ0	Alpha-actinin-4	*Actn4*	0.952	0.000	2	0.015	0.003	20	1.47
P47858	ATP-dependent 6-phosphofructokinase, muscle type	*Pfkm*	0.952	0.000	2	0.016	0.007	20	1.38
P13233	2,3-cyclic-nucleotide 3-phosphodiesterase	*Cnp*	0.940	0.012	2	0.016	0.006	28	0.52
P31977	Ezrin	*Ezr*	0.500	0.452	2	0.018	0.009	32	0.65
Q6PEC4	S-phase kinase-associated protein 1	*Skp1*	0.060	0.012	2	0.015	0.004	15	1.30
Q80U96	Exportin-1	*Xpo1*	0.060	0.012	2	0.019	0.011	20	1.81
Q6PDU1	Serine/arginine-rich splicing factor 2	*Srsf2*	0.048	0.000	2				1.49
Q812E9	Neuronal membrane glycoprotein M6-a	*Gpm6a*	0.048	0.000	2	0.017	0.006	18	0.21
Q5PPJ9	Endophilin-B2	*Sh3glb2*	0.952	0.000	1	0.016	0.005	16	1.58
Q9JLZ1	Glutaredoxin-3	*Glrx3*	0.952	0.000	1	0.016	0.005	26	1.51
P10536	Ras-related protein Rab-1B	*Rab1b*	0.952	0.000	1	0.016	0.004	34	0.63
Q923W4	Hepatoma-derived growth factor-related protein 3	*Hdgfrp3*	0.071	0.000	1				1.44
Q8K4V4	Sorting nexin-27	*Snx27*	0.071	0.000	1	0.017	0.006	5	1.33
Q62813	Limbic system-associated membrane protein	*Lsamp*	0.071	0.000	1	0.017	0.005	12	0.39
Q63617	Hypoxia up-regulated protein 1	*Hyou1*	0.071	0.000	1	0.018	0.007	40	0.38
P18297	Sepiapterin reductase	*Spr*	0.048	0.000	1				0.72
P05712	Ras-related protein Rab-2A	*Rab2a*	0.048	0.000	1				0.68
Q03346	Mitochondrial-processing peptidase subunit beta	*Pmpcb*	0.048	0.000	1	0.017	0.007	25	0.68
Q9JIX3	Bis(5-adenosyl)-triphosphatase	*Fhit*	0.048	0.000	1				0.67
P04762	Catalase	*Cat*	0.048	0.000	1				0.62
Q5EB77	Ras-related protein Rab-18	*Rab18*	0.048	0.000	1				0.59
P04041	Glutathione peroxidase 1	*Gpx1*	0.048	0.000	1				0.58
P11598	Protein disulfide-isomerase A3	*Pdia3*	0.048	0.000	1	0.019	0.007	10	0.55
Q4V7C6	GMP synthase [glutamine-hydrolyzing]	*Gmps*	0.048	0.000	1				1.31
Q62718	Neurotrimin	*Ntm*	0.048	0.000	1				0.49
O35346	Focal adhesion kinase 1	*Ptk2*	0.048	0.000	1	0.016	0.004	19	1.29
F1LMZ8	26S proteasome non-ATPase regulatory subunit 11	*Psmd11*	0.048	0.000	1	0.016	0.004	11	1.96
P31044	Phosphatidylethanolamine-binding protein 1	*Pebp1*	0.048	0.000	1				0.80
Q91ZN1	Coronin-1A	*Coro1a*	0.048	0.000	1	0.018	0.006	4	0.79
O35952	Hydroxyacylglutathione hydrolase, mitochondrial	*Hagh*	0.048	0.000	1				0.76
P22734	Catechol O-methyltransferase	*Comt*	0.048	0.000	1				0.76
O35964	Endophilin-A2	*Sh3gl1*	0.048	0.000	1	0.015	0.003	19	1.66
P85968	6-phosphogluconate dehydrogenase, decarboxylating	*Pgd*	0.048	0.000	1				0.73

## Data Availability

Data is contained within the article and Appendix A.

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
