# Peer review of "Prefrontal Cortex Cytosolic Proteome and Machine Learning-Based Predictors of Resilience toward Chronic Social Isolation in Rats"

_ijms, 2024, doi:10.3390/ijms25053026_

Round 1

Reviewer 1 Report

Comments and Suggestions for Authors

This article titled “Prefrontal cortex cytosolic proteome and machine learning-based predictors of resilience towards chronic social isolation in rats” by Filipovic et. al. describes an attempt to differentiate the resilience phenotype of rats subjected to chronic social isolation from rats susceptible to social isolation and control rats. Further, a group of protein classifiers is identified using machine learning approaches. Though the study is interesting, however, several concerns need to be addressed before the article is accepted for publication. Overall, the results need to be significantly improved.

Here are my concerns:

1. The authors can describe why only cytosolic proteome analysis was performed in this study. Was there any prior hypothesis regarding the cytosolic proteome?

2. Missing value imputation was mentioned for the feature selection analysis, however, similar details were missing for the differential proteome analysis.            

3. The results are poorly presented.

a. There is no summary of the proteomic analysis such as the number of proteins identified and quantified across the samples, and the number of proteins considered for the differential expression analysis etc.

b. Unsupervised clustering or PCA analysis could have been performed to visualize the group separation.

c. Volcano plot could have been shown for the differential expression analysis.

d. How do the authors explain the results comparing the CSIS-resilient group and control group? Why was a similar analysis not performed comparing CSIS-susceptible and control groups?

4. How was the class separation mentioned in line 229 performed?

5. How many proteins were used for the network analysis? Were only class separating proteins used or were all differentially expressed proteins used? If only class separating proteins were used, can the authors explain why?

6. How would the authors justify the applicability of machine learning approaches on such a poor sample size? How is the predictive classifier performance evaluated?

7. Finally, can the authors describe in the discussion a few potential scenarios where the predictive protein classifier would be useful and how it can be useful? How they can be used for shaping effective therapeutic strategies?

8. In Table 2, why do some of the proteins not have RF score, RF score SDEV, or RF times selected?

9. Line 263, “3.5.” needs to be removed. What does overlapping mean in line 264?

Author Response

Response to Reviewer 1 Comments

Reviewer 1

This article titled “Prefrontal cortex cytosolic proteome and machine learning-based predictors of resilience towards chronic social isolation in rats” by Filipovic et. al. describes an attempt to differentiate the resilience phenotype of rats subjected to chronic social isolation from rats susceptible to social isolation and control rats. Further, a group of protein classifiers is identified using machine learning approaches. Though the study is interesting, however, several concerns need to be addressed before the article is accepted for publication. Overall, the results need to be significantly improved.

Here are my concerns:

  1. The authors can describe why only cytosolic proteome analysis was performed in this study. Was there any prior hypothesis regarding the cytosolic proteome?

Response. We thank the reviewer for this suggestion. In the Introduction section we explained the reasons for performing cytosolic proteome analysis.  

  1. Missing value imputation was mentioned for the feature selection analysis, however, similar details were missing for the differential proteome analysis. 

Response: We thank the reviewer for these observations. Different proteome analysis was described in our previous study, Filipovic et al., Neurosci, 2022.

Filipović, D., Novak, B., Xiao, J., Yan, Y., Yeoh, K., Turck, C.W., 2022. Chronic Fluoxetine Treatment of Socially Isolated Rats Modulates Prefrontal Cortex Proteome. Neuroscience 501, 52–71. https://doi.org/10.1016/J.NEUROSCIENCE.2022.08.011

  1. The results are poorly presented.
  2. There is no summary of the proteomic analysis such as the number of proteins identified and quantified across the samples, and the number of proteins considered for the differential expression analysis etc.

Response: We thank the reviewer for these remarks. The summary of proteomic analysis has been added in the first paragraph of Comparative protein analysis of cytosolic-enriched fraction of rat PFC”, (303–310 in the revised manuscript).

  1. Unsupervised clustering or PCA analysis could have been performed to visualize the group separation.

Response: We appreciate the reviewer for this suggestion. We added the unsupervised PCA analysis to visualize the group separation. These additions can be found on lines 311-317 in the revised manuscript. Also, new Figs.3 A-B have been included that show PCA analysis.

  1. Volcano plot could have been shown for the differential expression analysis.

Response: According to the reviewer suggestion, we added a Volcano plot as a new Figs. 3 C-D.

  1. How do the authors explain the results comparing the CSIS-resilient group and control group? Why was a similar analysis not performed comparing CSIS-susceptible and control groups?

Response: We thank the reviewer for these questions. We added the sucrose preference test (SPT) results (Fig. 2). No significant differences in SP were observed in CSIS-resilient rats in terms of SP at the end of the 3rd and 6th weeks compared to baseline levels and also with control rats, which suggests that this resilient group did not have any depression-like phenotype. This is further confirmed by principal component analysis (PCA), where CSIS-resilient and control rats did not show separation (Figs. 3A–B). Besides, this result may be due to a limited threshold of 1.20 for the up-regulated proteins and 0.80 for the down-regulated proteins. We assume that these proteins were slightly up- or down-regulated in CSIS-resilient vs. control rats, indicating similar proteomic profiles between these groups, with only one upregulated protein.

The proteomic analysis of CSIS-susceptible and control groups was already published in Neurosciences, 2022. To avoid repeating the results in this study, we compared CSIS-resistant to CSIS-susceptible and control rats.

Filipović, D., Novak, B., Xiao, J., Yan, Y., Yeoh, K., Turck, C.W., 2022. Chronic Fluoxetine Treatment of Socially Isolated Rats Modulates Prefrontal Cortex Proteome. Neuroscience 501, 52–71. https://doi.org/10.1016/J.NEUROSCIENCE.2022.08.011

  1. How was the class separation mentioned in line 229 performed?

Response: We have added a more detailed description of how we define class-separating features with an example. These additions can be found on lines 245-250 in the revised manuscript. Below is an excerpt from the revised manuscript, with the added sentences highlighted in bold font:

“The described sequential feature selection (SFS) procedure is of limited use if the dataset contains features that enable perfect classification on their own. We define a feature as class-separating if its lowest value among all examples of one class is higher than its largest value in the other class. For example, the highest value for protein Q64537 in the CSIS-susceptible group was 10.153.000, while the lowest value for that same protein among the CSIS-resilient individuals was 17.733.000. In other words, the values for Q64537 were much lower for all individuals from the CSIS-susceptible group than for any individual from the CSIS-susceptible group. For such features, the SFS procedure stops after the first iteration, having achieved 100% validation accuracy by selecting a class-separating feature."

  1. How many proteins were used for the network analysis? Were only class separating proteins used or were all differentially expressed proteins used? If only class separating proteins were used, can the authors explain why?

Response: As predictive proteins, class separating proteins (n=45) that are differentially significant proteins between CSIS-resilient and CSIS-susceptible rats, were used for network analysis (Supplementary Table 3). With the use of STRING we wanted to define which proteome changes (cytosolic up/down regulated) have significant interactions among the changed class separating proteins as well as to define biological processes and molecular functions mostly affected by these changes. Enzyme code mapping was performed using STRING KEGG pathway software for class separating proteins. (Figure 3). With these predictive protein candidates, we could prospectively classify susceptibility or resilience to depressive phenotype and show different protein pathways involved in stress-coping strategies specific for the stress resilience phenotype.

  1. How would the authors justify the applicability of machine learning approaches on such a poor sample size? How is the predictive classifier performance evaluated?

Response: We agree with the reviewer that a larger sample size would be desirable. However, acquiring a large sample of this type is quite difficult due to ethical concerns and the difficulties with obtaining permits for sacrificing a large number of animals. 

Overfitting is a major concern when applying machine learning methods to smaller datasets, but we have employed all standard methods to avoid it. The sequential feature selection (SFS) analysis utilizes a very simple model (linear kernel SVM) and initially considers only a single feature. A model this simple is inherently robust towards overfitting. In later stages, as the number of features increases, the model gets more complex, but the validation accuracy stops improving (or hits the theoretical maximum of 100%) after just a few iterations (depending on the random seed that controls the train/test split). In most repetitions, the SFS procedure stops after just 2 iterations, resulting in models so simple that overfitting is unlikely, even with a sample of modest size like ours. Random Forests are more complex, but are not very susceptible to overfitting by design, due to the variance-reducing averaging that is inherent to the bagging method. To further reduce the dependence of our conclusions on chance (through random splitting of samples into the train and test sets in SVM, or through the random subspace method in RF), we repeated both the sequential feature selection and the RF-based feature importance scoring procedures 100 times each, with different random seeds. Finally, we identified those features which were consistently marked as highly informative by both procedures.

As for the evaluation of the classifier performance, we employed the standard cross-validation approach, where the data is split into folds, some of which are used to train the classifier, while others are used to evaluate its performance. As stated in the manuscript (line 231 in the revised manuscript), "We employed stratified 7-fold cross-validation for evaluating classifier performance." This results in balanced sets of 14 examples for training, and 2 examples (one per class) for evaluation, in each of the 7 iterations of the cross-validation procedure. The CSIS-resilient and CSIS-susceptible classes are very well separated, as is evident from the existence of class-separating features, and classifiers achieve 100% validation accuracies, as stated in lines 249-250 of the revised manuscript: “Identified potential predictive cytosolic proteins between CSIS-resilient vs. CSIS-susceptible groups by class separation make these groups linearly separable, whereby 100% validation accuracy is achieved by standard ML models.” However, we emphasize that our primary goal was not to train a classifier, but to identify informative predictors, which is why we have not included a more detailed evaluation of the trained classifiers in the manuscript.

  1. Finally, can the authors describe in the discussion a few potential scenarios where the predictive protein classifier would be useful and how it can be useful? How they can be used for shaping effective therapeutic strategies?

Response: We thank the reviewer for these suggestions. The proteins identified as informative in this research could easily be used to design a machine learning model for predicting if an individual is CSIS-resilient or CSIS-susceptible. This model would be trained by using the data collected in this research (or, preferably, a larger database), and could then be used to score a new individual as either CSIS-susceptible or CSIS-resilient. The score obtained from the machine-learning model could be used as one of the indicators for prioritizing resources towards treating more susceptible patients.”

  1. In Table 2, why do some of the proteins not have RF score, RF score SDEV, or RF times selected?

Response: Some features were identified as informative by the sequential feature selection (SFS) procedure, but not by the random forest method. The cells corresponding to the RF score, RF score SDEV and RF times selected for these features are blank. For example, P84076 was selected 5 times (out of 100) by the SFS procedure, albeit with a very low score of 0.052, but was not identified as informative by the RF in any of the 100 repetitions.

We have added the following sentence to the caption of Table 2 for clarification:

“Blank cells correspond to proteins not identified as informative by the RF-based procedure in any of the 100 repetitions.”

  1. Line 263, “3.5.” needs to be removed. What does overlapping mean in line 264?

Response: Based on the reviewer suggestion, we made the necessary corrections. Our intention was to emphasize that we are intersecting the results obtained by the two procedures (SFS and RF-based) for each protein individually. To eliminate a possible source of confusion, we have omitted the word ‘overlapping’.

Reviewer 2 Report

Comments and Suggestions for Authors

#1

The results presented by the authors are rather strange (Lines 224-228) "Differential proteomic analysis in the PFC cytosolic-enriched fractions revealed 170 downregulated and 218 upregulated proteins in the CSIS-resilient vs CSIS- susceptible rats (Supplementary Table 1). Compared with the control group, only one upregulated protein in CSIS-resilient rats was found (Supplementary Table 2)." From the data presented, it is clear that the CSIS-resilience and CSIS- susceptible groups are very strongly different from each other, but virtually no different from the control group. How to explain such a surprising correlation ?  This is a key question from the point of view of the value of the data presented and its potential biological significance. 

#2

Did the authors perform SPT (sucrose preferential test) for the control group ? If such a study was conducted, what would the classification of individuals into CSIS-resilience vs CSIS-susceptible look like ? The results of such studies would shed some light on the comment made in point 1.  

 #3

The description of the animal studies should include information on how many animals were used in the studies. The authors present data on 24 animals. However, it would be a remarkable coincidence to manage to randomly select eight individuals each belonging to the CSIS-resilience, CSIS-susceptible groups. 

#4

The description of the data and its presentation is very inconsistent. In the description of the methods (Lines 168-169 ) "Only proteins showing protein fold changes (FC) greater than or equal to 1.2 (f.c. ≥ 1.2) or less or equal to 0.80 (f.c. ≤ 0.80) ......". However, the description in lines 224-228 summarizes the entire range of FC, the data in Supplementary Table 1 also shows a list of all proteins. The same happens in the later parts of the manuscript. So how should we understand the description contained in the methods (lines 168-169) what does this description refer to ?

#5

Lines 230-232 "A panel of the top 15 cytosolic predictive proteins for discriminating CSIS-resilient vs. CSIS-susceptible rats is presented in Table 1. All proteins obtained by class separation are shown from the highest to the lowest score (Supplementary Table 3)." It is not clear from the description in the methods how this group of proteins was selected.  

#6

The description of the methods lacks information on whether technical repetitions were performed for each biological sample. If such repeats were performed, this should be stated in the methods description.

#7

In the Discussion section, the authors mainly refer to their own results (Filipović et al., 2020, Filipović et al., 2022 Filipović et al., 2023, Todorović et al., 2019), which could suggest the very niche nature of the research conducted.

Comments on the Quality of English Language

None

Author Response

Response to Reviewer 2 Comments

Reviewer 2

The results presented by the authors are rather strange (Lines 224-228) “Differential proteomic analysis in the PFC cytosolic-enriched fractions revealed 170 downregulated and 218 upregulated proteins in the CSIS-resilient vs CSIS- susceptible rats (Supplementary Table 1). Compared with the control group, only one upregulated protein in CSIS-resilient rats was found (Supplementary Table 2).” From the data presented, it is clear that the CSIS-resilience and CSIS- susceptible groups are very strongly different from each other, but virtually no different from the control group. How to explain such a surprising correlation ?  This is a key question from the point of view of the value of the data presented and its potential biological significance. 

Response: We thank the reviewer for this observation. In response to a reviewer’s observation, we added the sucrose preference test (SPT) results. As shown in Fig. 2, over a six-week period, CSIS-susceptible rats showed a significant reduction in SP at the end of 3 weeks (*p<0.05) compared to baseline level, and this difference remained statistically significant at the end of 6 weeks (***p<0.01). Furthermore, no significant differences in the SPT were observed in CSIS-resilient rats in terms of SP at the end of the 3rd and 6th weeks compared to baseline levels and also with control rats, which suggests that this resilient group did not have any depression-like phenotype, whereby the expression of proteins was similar between CSIS-resilient and control rats, with only one upregulated protein. This is further confirmed by principal component analysis (PCA), which showed a significant difference in the proteomic features of resilient and susceptible rats to CSIS, while resilient and control rats did not show any difference between them (Figs 3 A-B), which may indicate similar proteomic profiles between these groups.

Besides, this result may be due to a limited threshold of 1.20 for the up- and 0.80 for the down-regulated proteins. We assume these proteins were slightly up/down-regulated in CSIS-resilient vs. control rats.

#2

Did the authors perform SPT (sucrose preferential test) for the control group? If such a study was conducted, what would the classification of individuals into CSIS-resilience vs CSIS-susceptible look like? The results of such studies would shed some light on the comment made in point 1.  

Response: We thank the reviewer for this important suggestion. New Fig. 2 which represents SPT for rat classification into CSIS-resilient and CSIS-susceptible groups, has been added in the revised manuscript with the aim of providing a clarification of the remarks from the first point.

 #3

The description of the animal studies should include information on how many animals were used in the studies. The authors present data on 24 animals. However, it would be a remarkable coincidence to manage to randomly select eight individuals, each belonging to the CSIS-resilience, CSIS-susceptible groups. 

Response: We thank the reviewer for this observation. This current study was a part of research that examined altered proteomic components and biochemical pathways and processes that may underlie the development of a pathological phenotype or not, as well as the response to a pharmacological treatment with fluoxetine (Flx), in an animal model of major depressive disorders based on the exposure of rats to a paradigm of chronic social isolation (CSIS).

The results of sucrose preference were used as the criteria for rat’s selection for resilient or susceptible towards CSIS stress which was done prior to the start (0 week-baseline), at the end of the 3rd and the 6th weeks of the experiment. The experiment started with 50 rats.16 out of 20 animals in Control or Control+Flx, and 24 out of 30 in CSIS-susceptible, CSIS + Flx and CSIS-resilient groups, were used for the experiments. Based on behavior testing data, rats were segregated to: CSIS-susceptible (an decrease in SP of > 30% at the end of the 6th weeks of testing compared to baseline, n=12); CSIS resilient (their SP at the end of the 6th weeks was no significant difference with baseline or controls n=8); rats responding effectively to Flx treatment, i.e., CSIS + Flx group (an increase in SP > 30% at the end of 6th n=6); Flx resilient with no reversed reduction in SP at the end of 6th week (n=4). Hence, all animals were not included in the current study. From control rats (n=20), 2 rats from Control group and 2 rats from Control + Flx group were excluded, since they didn’t show relevant behavior. Considering that we had 8 rats in control and CSIS-resilient group, the final number of animals per group in this study was 8. The remaining animals (CSIS-susceptible) were used for a metabolomic analysis pilot study.

#4

The description of the data and its presentation is very inconsistent. In the description of the methods (Lines 168-169 ) "Only proteins showing protein fold changes (FC) greater than or equal to 1.2 (f.c. ≥ 1.2) or less or equal to 0.80 (f.c. ≤ 0.80) ......". However, the description in lines 224-228 summarizes the entire range of FC, the data in Supplementary Table 1 also shows a list of all proteins. The same happens in the later parts of the manuscript. So how should we understand the description contained in the methods (lines 168-169) what does this description refer to?

Response: We thank you for this remark. In the original manuscript description lines 224-228 summarizes CSIS-resilient vs. CSIS-susceptible proteins with false discovery rate (FDR)-adjusted p-values of < 0.05 and fold change (FC) greater than or equal to 1.2 (f.c. ≥ 1.2) or less or equal to 0.80 (f.c. ≤ 0.80) as statistically significant, which are represented in Supplementary Table 1. According to the 1st reviewer’s suggestion, in the revised manuscript we included a summary of the proteomic analysis, such as the number of proteins identified and quantified across the samples, and the number of proteins considered for the differential expression analysis.

#5

Lines 230-232 "A panel of the top 15 cytosolic predictive proteins for discriminating CSIS-resilient vs. CSIS-susceptible rats is presented in Table 1. All proteins obtained by class separation are shown from the highest to the lowest score (Supplementary Table 3)." It is not clear from the description in the methods how this group of proteins was selected.  

Response: We have added a more detailed description of how we define class-separating features with an example. These additions can be found on lines 243-249 in the revised manuscript. Below is an excerpt from the revised manuscript, with the new sentences highlighted in bold font:

“The described sequential feature selection (SFS) procedure is of limited use if the dataset contains features that enable perfect classification on their own. We define a feature as class-separating if its lowest value among all examples of one class is higher than its largest value in the other class. For example, the highest value for protein Q64537 in the CSIS-susceptible group was 10.153.000, while the lowest value for that same protein among the CSIS-resilient individuals was 17.733.000. In other words, the values for Q64537 were much lower for all individuals from the CSIS-susceptible group than for any individual from the CSIS-susceptible group. For such such features, the SFS procedure stops after the first iteration, having achieved 100% validation accuracy by selecting a class-separating feature.”

#6

The description of the methods lacks information on whether technical repetitions were performed for each biological sample. If such repeats were performed, this should be stated in the methods description.

Response: Technical repetitions were not performed for each biological sample.

#7

In the Discussion section, the authors mainly refer to their own results (Filipović et al., 2020, Filipović et al., 2022 Filipović et al., 2023, Todorović et al., 2019), which could suggest the very niche nature of the research conducted.

Response: We agree with the reviewer comment. A more profound discussion on this issue was included in the revised manuscript in the Discussion section. Also, new references have been added.

Reviewer 3 Report

Comments and Suggestions for Authors

The current manuscript focuses on the study of chronic social isolation (CSIS) in rats through the use of proteomic methods. Specifically, the authors isolated proteins from rats that were treated differently for CSIS and quantified them. While the manuscript is well-written, the experimental ambiguity makes it challenging to draw conclusions. The proteomic comparison between the control, CSIS-resilient, and CSIS-susceptible groups is particularly ambiguous. The machine learning methods as stated in the title are not mentioned in the manuscript. To improve the manuscript, the following points could be considered.

1.     The structure of the study is not sufficiently apparent. The process of selecting rats randomly for the control and CSIS groups may have been subject to bias due to the failure to take into account and evaluate any pre-existing conditions. The number of animals included in each group needs to be specified in Figure 1.

2.     The authors need to describe the protein isolation from the brain tissues in brief. How much protein was isolated and how much was digested from each sample? What minor modifications were made in the filter-aided sample preparation method?

3.     The LC and MS methods do not clearly state the right experimental conditions. “Peptide ions with charge states of 2+ to 4+ and above were selected for fragmentation” This is an ambiguous statement. What is the highest charge state of the peptide that was fragmented?

4.     “Only proteins showing protein fold changes (FC) greater than or equal to 1.2 (f.c. ≥ 1.2) or less or equal to 0.80 (f.c. ≤ 0.80) and a p value < 0.05 and FDR < 0.05 were considered differentially expressed.” This is not a clear statement. The authors need to show the volcano plot to validate this statement.

5.     What is class-separating proteins? How were they selected? The FC in Table 1 is less than 1 which shows these proteins are neither up or down-regulated. The proteins selected for network analysis from the class-separated table do not appear to be differentially expressed proteins.

6.     The comparative protein analysis of cytosolic-enriched fraction should be discussed in detail as this is highly important in determining the up and down-regulated proteins in each condition.

Minor comments:

1.     The abstract needs revision and state the methods used in this study.

2.     The methods section needs to state the chemical and reagents used. Also, a detailed experimental method is needed to reproduce the results.

3.     Describe and discuss the pathway analysis in detail.

4.     Conclusions are not clear and are not supported by confident results.

Comments on the Quality of English Language

The manuscript needs thorough revision for grammar and spelling. 

Author Response

Response to Reviewer 3 Comments

Reviewer 3

The current manuscript focuses on the study of chronic social isolation (CSIS) in rats through the use of proteomic methods. Specifically, the authors isolated proteins from rats that were treated differently for CSIS and quantified them. While the manuscript is well-written, the experimental ambiguity makes it challenging to draw conclusions. The proteomic comparison between the control, CSIS-resilient, and CSIS-susceptible groups is particularly ambiguous. The machine learning methods as stated in the title are not mentioned in the manuscript. To improve the manuscript, the following points could be considered.

  1. The structure of the study is not sufficiently apparent. The process of selecting rats randomly for the control and CSIS groups may have been subject to bias due to the failure to take into account and evaluate any pre-existing conditions. The number of animals included in each group needs to be specified in Figure 1.

Response: We thank the reviewer for this observation. This current study was part of an investigation that examined altered proteomic components and biochemical pathways and processes that may underlie the development of a pathological phenotype or not, as well as the response to a pharmacological treatment with fluoxetine (Flx), in an animal model of major depressive disorders based on the exposure of rats to a paradigm of chronic social isolation (CSIS).

The results of sucrose preference were used as the criteria for rat’s selection on resilient or susceptible towards CSIS stress which was done prior to the start (0 week-baseline), at the end of the 3rd and the 6th weeks of the experiment. The experiment started with 50 rats.16 out of 20 animals in Control or Control+Flx, and 24 out of 30 in CSIS-susceptible, CSIS + Flx and CSIS-resilient groups, were used for the experiments. Based on behavior testing data rats were segregated to: CSIS-susceptible (an decrease in SP of > 30% at the end of the 6th weeks of testing compared to baseline, n=12); CSIS resilient (their SP at the end of the 6th weeks was no significant difference with baseline or controls n=8); rats responding effectively to Flx treatment, i.e., CSIS + Flx group (an increase in SP > 30% at the end of 6th n=6); Flx resilient with no reversed reduction in SP at the end of 6th week (n=4). Hence, all animals were not included in the current study. From control rats (n=20), 2 rats from Control group and 2 rats from Control + Flx group were excluded, since they did not show relevant behavior. Considering that we had 8 rats in control and CSIS-resilient group, the final number of animals per group in this study was 8. The remaining animals (CSIS-susceptible) were used for a metabolomic analysis pilot study.

  1. The authors need to describe the protein isolation from the brain tissues in brief. How much protein was isolated and how much was digested from each sample? What minor modifications were made in the filter-aided sample preparation method?

Response: We thank the reviewer for this suggestion. We added the protocol for enzymatic digestion of proteins in solution. After the isolation of cytosolic-enriched fractions, the concentration of proteins in the samples was 2 - 4 mg/ml. For digestion of proteins, we used 20 µg of the proteins from the cytosolic-enriched fraction of each sample.

Regarding the modification of used protocol for enzymatic digestion of proteins in solution in our study compared with original protocol Wiśniewski et al., 2009, we used:

  1. filter unit Microcon YM-30, instead of Microcon-10kDaCentrifugal Filter;
  2. 10 mM Tris/HCl buffer for homogenisation of PFC samples without DTT to measure the protein concentration with Lowry and BCA assays and then we incubated with 5mM DTT in UA (8M urea in 0.05M Tris/HCl pH 8.5) 1h 37°C, while in original protocol, SDT-buffer with 0.1M DTT was used and samples were heated for 3min at 95°C;
  3. 100mM TEAB (Tetraethylammoniumbromide) instead of ABC (0.05M NH4HCO3) and 0.5M NaCl was used.

  1. The LC and MS methods do not clearly state the right experimental conditions. “Peptide ions with charge states of 2+ to 4+ and above were selected for fragmentation” This is an ambiguous statement. What is the highest charge state of the peptide that was fragmented?

Response: We thank the reviewer for this observation. We added that peptide ions with charge states of 2+ to 4+ (above is deleted in the revised manuscript) were selected for fragmentation (line 194-195 in the revised manuscript). The highest charge state of the peptide that was fragmented is +4.

Weckmann, K., Deery, M.J., Howard, J.A., Feret, R., Asara, J.M., Dethloff, F., Filiou, M.D., Iannace, J., Labermaier, C., Maccarrone, G., Webhofer, C., Teplytska, L., Lilley, K., Müller, M.B., Turck, C.W., 2017. Ketamine’s antidepressant effect is mediated by energy metabolism and antioxidant defense system. Sci. Reports 2017 71 7, 1–11. https://doi.org/10.1038/s41598-017-16183-x

  1. “Only proteins showing protein fold changes (FC) greater than or equal to 1.2 (f.c. ≥ 1.2) or less or equal to 0.80 (f.c. ≤ 0.80) and a p value < 0.05 and FDR < 0.05 were considered differentially expressed.” This is not a clear statement. The authors need to show the volcano plot to validate this statement.

Response: We thank the reviewer for this important suggestion, and we added a volcano plot to Fig. 3. C-D

  1. What is class-separating proteins? How were they selected? The FC in Table 1 is less than 1, which shows these proteins are neither up- or down-regulated. The proteins selected for network analysis from the class-separated table do not appear to be differentially expressed proteins.

Response: We have added a more detailed description of how we define class-separating features with an example. These additions can be found on lines 243-249 in the revised manuscript. Below is an excerpt from the revised manuscript, with the added sentences highlighted in bold font:

“The described sequential feature selection (SFS) procedure is of limited use if the dataset contains features that enable perfect classification on their own. We define a feature as class-separating if its lowest value among all examples of one class is higher than its largest value in the other class. For example, the highest value for protein Q64537 in the CSIS-susceptible group was 10.153.000, while the lowest value for that same protein among the CSIS-resilient individuals was 17.733.000. In other words, the values for Q64537 were much lower for all individuals from the CSIS-susceptible group than for any individual from the CSIS-susceptible group. For such features, the SFS procedure stops after the first iteration, having achieved 100% validation accuracy by selecting a class-separating feature."

  1. The comparative protein analysis of cytosolic-enriched fraction should be discussed in detail as this is highly important in determining the up and down-regulated proteins in each condition.

Response: We thank the reviewer for this suggestion. New information has been added (lines 303–310 in the revised manuscript).

Minor comments:

  1. The abstract needs revision and state the methods used in this study.

Response: As the reviewer requested, we have added the methods used in the study to the Abstract.

  1. The methods section needs to state the chemical and reagents used. Also, a detailed experimental method is needed to reproduce the results.

Response: We thank the reviewer for this comment. New information has been added in the Methods section on this issue along with new references.

  1. Describe and discuss the pathway analysis in detail.

Response: As reviewer suggested, we discussed pathway analysis in details (lines 360–370) in the revised manuscript)

  1. Conclusions are not clear and are not supported by confident results.

Response: We thank the reviewer for this suggestion. We have more extensively stated the conclusions of our study.

Round 2

Reviewer 3 Report

Comments and Suggestions for Authors

The authors revised the manuscript as per suggestions and looks much improved.